# Emphasis Learning, Features Repetition in Width Instead of Length to Improve Classification Performance: Case Study—Alzheimer’s Disease Diagnosis

**DOI:** 10.3390/s20030941

**Published:** 2020-02-10

**Authors:** Hamid Akramifard, MohammadAli Balafar, SeyedNaser Razavi, Abd Rahman Ramli

**Affiliations:** 1. Faculty of Electrical and Computer Engineering, University of Tabriz, East Azerbaijan, Tabriz 51666-16471, Iran; hamid.akramifard@tabrizu.ac.ir (H.A.); n.razavi@tabrizu.ac.ir (S.R.); 2. Department of Computer and Communication Systems Engineering, University Putra Malaysia, UPM-Serdang 43400, Malaysia; arr@eng.upm.edu.my

**Keywords:** Alzheimer’s disease, Emphasis Learning, multi-modal classification, SVM, PCA

## Abstract

In the past decade, many studies have been conducted to advance computer-aided systems for Alzheimer’s disease (AD) diagnosis. Most of them have recently developed systems concentrated on extracting and combining features from MRI, PET, and CSF. For the most part, they have obtained very high performance. However, improving the performance of a classification problem is complicated, specifically when the model’s accuracy or other performance measurements are higher than 90%. In this study, a novel methodology is proposed to address this problem, specifically in Alzheimer’s disease diagnosis classification. This methodology is the first of its kind in the literature, based on the notion of replication on the feature space instead of the traditional sample space. Briefly, the main steps of the proposed method include extracting, embedding, and exploring the best subset of features. For feature extraction, we adopt VBM-SPM; for embedding features, a concatenation strategy is used on the features to ultimately create one feature vector for each subject. Principal component analysis is applied to extract new features, forming a low-dimensional compact space. A novel process is applied by replicating selected components, assessing the classification model, and repeating the replication until performance divergence or convergence. The proposed method aims to explore most significant features and highest-preforming model at the same time, to classify normal subjects from AD and mild cognitive impairment (MCI) patients. In each epoch, a small subset of candidate features is assessed by support vector machine (SVM) classifier. This repeating procedure is continued until the highest performance is achieved. Experimental results reveal the highest performance reported in the literature for this specific classification problem. We obtained a model with accuracies of 98.81%, 81.61%, and 81.40% for AD vs. normal control (NC), MCI vs. NC, and AD vs. MCI classification, respectively.

## 1. Introduction

Alzheimer’s disease (AD) can be described by cognitive and memory dysfunctions. This in turn is actually the major cause of dementia in older adults. Moreover, AD has been identified as one of the main causes of death in the United States [1]. The early diagnosis and prognosis of AD are important because of limitations in treatment time.

In this area, many biomedical imaging techniques for the early detection of AD are well developed and employed by researchers, including MRI [2,3,4], PET [5,6], and other features like CSF [7], and the Mini-Mental State Examination (MMSE) [8].

To develop automated procedures based on the above techniques to detect brain atrophy, in the initial stages of AD, regions including the entorhinal cortex, the hippocampus, lateral and inferior temporal structures, and anterior and posterior cingulate cortex [9,10,11,12] have been reported. Previous works have dealt with the construction of computer-aided diagnosis (CAD) systems. Almost all of these CAD systems are based on machine learning techniques and have three main steps: data pre-processing, feature extraction, and classification. The pre-processing procedure sets different images from different subjects, with brains of different sizes and shapes, at a comparable condition and cleans and imputes missing data from the obtained data (if any). In the second step, a feature extraction algorithm converts the input data into small vectors [13]. The classifier determines if the vectors are more similar to mild cognitive impairment (MCI) patient vectors, to AD patient vectors, or to normal control (NC) vectors. 

To use these CAD systems, the metrics of entorhinal cortex have been used in AD diagnosis [14]. Automatic hippocampal volume measurement methods have almost equal results [15,16]. Hippocampal volumes and entorhinal cortex metrics seem to be equally accurate in distinguishing between AD patients and NC subjects [17]. Different techniques, such as principal component analysis (PCA), artificial neural networks (ANNs), fuzzy neural networks (FNNs), partial least square (PLS), and support vector machine (SVM), have been used in the development of these CADs. 

These brain-observing techniques using machine learning can provide tools to overcome brain dysfunction problems. These combined techniques can use different modalities including MRI, PET, and other neurological data to diagnose AD/MCI patients from healthy people [18,19,20,21]. In [22] 50 MRI images from the OASIS dataset were used for characterization of MRIs of brains affected with Alzheimer’s disease by fractal descriptors. Additionally, [23] used MRI images from the Alzheimer’s Diseases Neuroimaging Initiative (ADNI) dataset for distinguishing AD from NC. They reported complete performance (100% accuracy) in distinguishing between the two groups. [18] reported a multiple classification using transfer learning on AD, while [24] classified AD vs. NC with a great rate of accuracy using only MRI data. [25] classified progressive MCI vs. static MCI using combined MRI, APOe4 genetic, and cognitive measures include and APOe4 genotyping. 

In this area, the feature extraction and feature combination are often performed independently. As investigated in the previous studies, there are inherent relations between the modalities of MRI and PET [26]. Thus, finding the shared feature representation that combines the complementary information from different modalities (e.g., PET, MRI, and CSF) is useful to enhance the discrimination of AD and MCI patients from NC subjects.

There are some features among the described data that can help us to better diagnose AD. We use PCA for dimensionality reduction and recognition of the possibly most efficient features of the data to enhance classification potential. Feature representation using PCA reduces processing resources usage, in addition to enhancing the classification accuracy. The steps of the proposed approach can be summarized as follows:(1)Feature extraction from MRI images and other data sources (from the ADNI dataset).(2)Concatenation of all the features.(3)Preparation of data sets and refining the data.(4)Dimension reduction using PCA.(5)Repeating data in vectors of step 4 until achieving the highest classification performance.

Briefly, the main contributions of this study are as follows:A novel method named Emphasis Learning is proposed for improving classification performance.The proposed method is successfully adapted for the diagnosis and prognosis of AD patients and distinguishing them from normal subjects.

This methodology is the first of its kind in the literature, and works based on the notion of replication on the feature space instead of the traditional sample space. PCA was employed only asan expert feature engineer to extract high-variance features. The proposed method achieved about 99% accuracy in classifying normal subjects from AD patients. This result outperformed all of the current literature results in terms of accuracy in classifying AD patients from normal subjects.

In the following sections, the data extraction and data source are presented. Then, pre-processing, feature reduction and increment, and classification methods are explained. In the following section, the experimental results are provided. A discussion of the results and conclusions are presented in the final sections.

## 2. Materials and Methods

In this method, we emphasized the strongest and most influential features. The main idea of this article is rooted in the fact that when a person’s good and outstanding features are emphasized, those features along with the person himself improve, and his performance improves as well. In other words, these features improve themselves; and the better and more precise these features, the more effective they can be. To put this another way, learning can result either when good features in small quantities repeat several times, or when such positive features repeat few times. For instance, which of the two following ways would result in higher learning rates? When a teacher solves a problem with two solutions and repeats these many times? Or, when she uses many solutions while repeating each just a few times? The answer is “both”. However, for some problems the former works better, and for some, the latter. The same is true in computer applications, and deeper and broader learning takes place when there is the possibility of repeating both approaches.

It is obvious that trying to increase the accuracy of a classifier that is fine-tuned is a challenge. For example, usually, increasing the accuracy from 80% to 85% is less complex and needs lower costs and computational burden than increasing it from 95% to 96% (if possible). In this study, a method is put forward in order to make this possible and to reduce the costs and computational burden to a great extent. However, when the computed performance using the main data is low, or when utilizing the extracted features from the dimension reduction does not cause a change in the model’s performance (i.e., if no outstanding feature is achieved), repeating these features may not be that influential in the tuning precision.

### 2.1. Characteristics of Subjects

We only used baseline MRI and PET image data, as well as CSF data acquired from 156 AD patients, 338 MCI patients, and 211 NC subjects from the ADNI dataset. Table 1 shows patients’ demographic information. All the data were acquired in May 2017. Mini-Mental State Examination (MMSE) scores were added to the extracted data from the ADNI database. MMSE scores have three ranges that are defined as follows: (1) scores between 24 and 30 which represent healthy people; (2) scores between 20 and 24 that represent MCI subjects; (3) scores between 13 and 20 that represent moderate dementia subjects.

### 2.2. MRI and PET Images and CSF Data

The MRI images were in Neuroimaging Informatics Technology Initiative (NIfTI) format. These MRI images were pre-processed for spatial distortion correction. Collection of CSF data was done in the morning after an overnight use of 20- or 24-gauge spinal needle. The FluoroDeoxyGlucose-Positron Emission Tomography (FDG-PET) images were average values obtained from 30–60 min post injection; their voxel size was converted to standard and smoothed to 8 mm full width at half of maximum resolution.

#### 2.2.1. MRI Acquisition Parameters

In the multiple ADNI sites, multiple machines (e.g., Siemens, Philips, and GE Medical scanners) are used. Standard protocol was developed to evaluate 3D T1-weighted sequences for morphometric analyses [27]. Structural brain MRI scans were acquired using 1.5 T and 3 T MRI scanners. Most of the 1.5 T MRIs were obtained from GE Medical scanners, and most of the 3 T MRIs were acquired from Siemens machines.

In the 1.5 T protocol, each subject experienced 2 × 1.5 T T1-weighted MRI by 3D sagittal volumetric magnetization-prepared rapid gradient echo (MP-RAGE) sequence. The repetition time of typical 1.5 T acquisition was 2400 ms, and the inversion time was 1000 ms. Flipping and field of view were 8° and 24 cm, respectively. Dimensionality of MRIs was 256 × 256 × 170, and the voxel size was 1.25 × 1.25 × 1.2 mm^3^.

For 3 T scans, repetition time and inversion time were 2300 and 900 ms, respectively. Flipping angle and field of view were 8° and 26 cm. Dimensionality of MRIs was 256 × 256 × 170, with voxel size of 1.0 × 1.0 × 1.2 mm^3^.

For modern systems, the scan time at 1.5 T is 7.7 min, and for 3 T systems it is 9.3 min. This usually happens because of the difference between vulnerability artifacts, spin relaxation, and chemical shift properties in 1.5 T and 3 T systems.

Figure 1 shows a sample MRI imaging of an NC subject and an AD patient. The figure demonstrates decreased gray matter (GM) volume in the AD patient compared to the normal control.

#### 2.2.2. Pre-Processing of MRI Images

Spatial parametric mapping (SPM) software was used for pre-processing [28]. SPM was used for realignment, smoothing, spatial normalization, and feature extraction from MRI regions of interest (ROIs). The pre-processing steps using VBM8 tools were as below:Check that image format is in a suitable condition using SPM tools.Segment the images to identify gray matter and white matter (WM) and wrap GM to the segmented image to Montreal Neurological Institute (MNI) space using the SPM tools.Estimate deformations to best align the images to each other and create templates by registering the imported images with their average, iteratively using DARTEL tools of SPM.Generate spatially normalized and smoothed GM images normalized to MNI space. Using the estimated deformations by the DARTEL tools of SPM, generate smoothed/modulated wrapped GM and WM images.

Note that using CAT12 for MRI segmentation and feature extraction can also obtain promising results, as reported by Farokhian et al. [29]. This study demonstrated better performance using CAT12 over VBM8 tools.

Data cleansing and selection were done in the pre-processing step. In the second step (feature extraction), the input data were converted into small vectors [13]. The classification algorithm determined whether the vectors are more similar to an MCI/AD patient or to a normal control (NC) subject.

### 2.3. Feature Extraction

To extract the features of all the images, we adopted standard procedures of anterior commissure (AC)–posterior commissure (PC) correction, skull-stripping, and cerebellum removal for pre-processing and preparing. MIPAV software was used for AC–PC correction. We segmented structural MRI images into WM, GM, and CSF images (in the literature, GM matter has been widely used in AD diagnosis, and most of the brain structural MRI studies in AD focused on only gray matter abnormalities [30]). Then, to extract ROI features of all the images, we used Voxel Based Morphometry tools of Spatial Parametric Mapping (VBM-SPM) to extract GM features as well. After that, volumetric changes in specific regions such as entorhinal cortex, hippocampus, and temporal and parietal lobes were used. For each ROI, a mask was made using WFU Pick Atlas tools (https://www.nitrc.org/projects/wfu_pickatlas/).

The PET images were aligned to the corresponding MR images strictly in ADNI. The recognized areas including MRI gray matter tissue volume, average voxel values, and average of PET voxel values (that were downloaded from the ADNI database) were used as features. In the literature, these features have been used for AD/MCI diagnosis [31,32,33,34]. Three CSF biomarkers (i.e., Ab42, t-tau, and p-tau) were also used in making the feature set. Therefore, 144 features were used to form the final feature set consisting of 132 Voxel values and Volume of MRIs, 1 MMSE score, 4 pieces of personal information, 3 CSF biomarkers, and 4 PET voxel values (since we could not access PET images in the ADNI database, we used only four PET voxel values that were extracted and uploaded to the ADNI database). Finally, the vectors of the extracted features were normalized by applying natural logarithm. K-fold cross validation method was used for testing and evaluation.

To parcellate the brain, Automatic Anatomical Labeling (AAL) atlas (http://www.gin.cnrs.fr/en/tools/aal/) (Figure 2) was used, as proposed by [35]. In this atlas, the brain is parcellated into 90 cerebral regions and 26 cerebellar regions. The sixteen most effective regions of the brain for this work are respectively the left amygdala, inferior temporal gyrus, left middle frontal gyrus, left inferior temporal gyrus, right amygdala, left middle temporal gyrus, left middle temporal gyrus, left supramarginal gyrus, left middle frontal gyrus, left inferior frontal gyrus, left hippocampus, left angular gyrus, left superior frontal gyrus, right supramarginal gyrus, right hippocampus, and right parahippocampal gyrus. We used both voxel values and volumes of these regions in this study.

## 3. Classification Methods

For diagnosis and prognosis of AD and MCI, some classification algorithms are common and some algorithms play supportive roles. Among them, SVM and PCA are the most used.

### 3.1. Feature Reduction Method

One of the most common linear techniques for data dimension reduction is PCA. PCA was introduced by Karl Pearson in [36]. It maps the data to a lower dimension while maintaining the data’s variance. To use this method, the covariance matrix of the data and the eigenvectors on this matrix must be computed. The eigenvectors from the largest eigenvalues (i.e., the principal components) reconstruct the highest variance of the primary data. The first few eigenvectors often have the most information of the primary data. Hence, the process yields a smaller number of eigenvectors, and there may be some associated data loss. However, the most important variances should be retained by the remaining eigenvectors. Figure 3 shows the eigenvectors of a dataset.

PCA has the advantages of reducing the required storage space and computation time eliminating redundant features. Some associated disadvantages include a reduction in some of the original data’s information, its failure when mean and covariance are not sufficient to define the data, and uncertainty in the number of principal components required to retain the data’s information.

### 3.2. Increasing Dimensions of Data to Achieve Better Classification Results

The main idea of this paper is to repeat the most efficient features in classification. Theoretically, increasing the dimensions of data can sometimes yield better and sometimes worse classification results; but what if we find and repeat good data features to make a classification model? Our experiment shows that this theory worked excellently! For the diagnosis of Alzheimer’s Disease, we tested this method after extracting the best features of the data set using PCA, conducted a dimension reduction, and repeated these new features. We repeated these features as input data for the classification algorithm (here SVM) until this action had no further positive effect on classification performance or did not reduce it. Figure 4 shows the proposed method diagram. The performance of the proposed model increased in some cases after applying PCA, but in many other cases it might lead to performance loss. Because using more PCs can lead to performance loss, we experimentally found that using only 25 gave the best results.

### 3.3. SVM

One binary classification method that is successfully used in many domains is SVM [3,37,38,39]. The classification efficiency of SVM in training very-high-dimensional data has been proven [40,41]. Moreover, SVM has been applied to voice activity detection, pattern recognition, classification, and regression analysis [42,43]. It is used to separate a set of training data with a hyperplane that is maximally distant from the two classes. SVM is the most common and efficient classifier in binary classification. Here, SVM was used to distinguish between AD and MCI patients and NC subjects, pairwise. We employed only the training samples to adjust the hyperparameters of the SVM, based on which the best model was selected. Using the proposed method, we could automatically select the best model among other trained models due to the achieved performance rates.

### 3.4. Data Normalization vs. Data Standardization

Normalization maps values into a range of [0,1] and it is effective in the applications that require positive values. In this study, we used a normalization method. Equation (1) shows the normalization formula:(1)$$X_{norm}=\frac{X-X_{min}}{X_{max}-X_{min}},$$
where ***X**_norm_* is the normalized input data. The problem with normalization is outlier elimination.

On the other hand, standardization maps original data to have a mean of 0, and is recommended in some cases. Equation (2) demonstrates standardization:(2)$$X_{std}=\frac{X-X_{mean}}{X_{dev}},$$
where ***X****_std_* implies standardized data, *X_mean_* is the mean of the original data, and *X_dev_* is the standard deviation of the data.

### 3.5. Evaluation Criteria

Accuracy is a well-known evaluation measure for classification methods. Using accuracy, we computed the correctly classified samples and all samples’ ratio. Two other common evaluation metrics are sensitivity and specificity. The receiver operating characteristic (ROC) and the area under the curve (AUC) are other performance parameters for diagnosis procedures. The positive predictive value (PPV) and negative predictive value (NPV) are widely used measures to describe the performance of a classifier. The accuracy, sensitivity, specificity, PPV, and NPV are defined in the following equations, respectively:(3)$$Acc=\frac{TP+TN}{TP+TN+FP+FN}$$
(4)$$Sen=\frac{TP}{TP+FN}$$
(5)$$Spec=\frac{TN}{TN+FP}$$
(6)$$PPV=\frac{TP}{TP+FP}$$
(7)$$NPV=\frac{TN}{TN+FN}$$
where TP is the number of true positives (number correctly classified as patients); TN is the number of true negatives (number correctly classified as non-patients); FP is the number of false positives (number of non-patients wrongly classified as patients); FN is the number of false negatives (number of patients wrongly classified as non-patients). 

We used the sensitivity and specificity to evaluate the rate of true positives or negatives (i.e., the ratio of correctly classified AD or MCI patients or NC subjects and total subjects). These measures show the method’s detection power between AD, MCI, and NC. Here, these metrics were measured using K-fold cross validation (with k = 10). Using this method, 10 selected sets of AD, MCI, and NC were sampled randomly—one set for testing and nine for training the classifier. This was be done for all 10 sets, and the average of the evaluation parameters was be used to show the performance of the classification method. In this article, we repeated the K-fold method 100 times and the average of averages was used to represent the method’s performance.

The steps of the algorithm are as follows (Algorithm 1):
**Algorithm 1.** The steps of the algorithm of the method.Feature extractionMF <- MRI ROI featuresPF <- PET ROI featuresCD <- CSF dataPI <- Personal informationMS <- MMSE ScoreConcatenated Features <- [MF, PF, CD, PI, MS]Imputaded data <- Imputation (Concatenated Features)data <- ln(Imputaded data) ‘data normalization using Natural Logarithm’Normalized data <- (data–min(data)/(max(data)–min(data))) ‘data normalization (Xnorm)’Reduced data <- PCA (Normalized data)Data <- [Reduced data]Diagnosis <- SVM(Data) ‘classificaion using SVM’Data <- [Data, Reduced data]Go to (8) until no further performance improvements are obtainedSelect the best performance and Finish

The third step was concerned with missing values, because there are many missing values in PET and CSF data. The missing values were replaced by the average (mean) of the existing items for all subjects as suggested by [44]. This method was successfully applied to fill the missing values and we saw the performance gain after using that. Note that PET and CSF examinations have difficult processes. For this reason, some of the ADNI subjects were avoid from these experiments. It should be said that none of MMSE, MRI, or demographic data had missing values so all of them were used in our experimentations.

## 4. Experimental Results

Here, we evaluate our proposed method’s efficiency. This was done for three binary classification problems: AD vs. NC, MCI vs. NC, and AD vs. MCI, and a 10-fold cross validation method was used for evaluation purposes. In the 10-fold cross validation, the dataset was randomly partitioned into 10 subsets, each including one-tenth of the total dataset. Nine subsets were used for training goals and the remaining one for testing. We did this for all subsets.

### Classification Results

In order to represent the performance of the proposed method, we present the classification results obtained from the SVM classification algorithm by 10-fold cross validation. Table 2 shows the mean accuracy, sensitivity, specificity, positive predictive value, negative predictive value, and area under the curve for different numbers of the repeated features tested on the proposed method on three binary classes. As can be seen, by repeating features five times, the proposed method showed the best accuracy rates of 98.81%, 81.61%, and 81.40% in classifying NC vs. AD, MCI vs. AD, and NC vs. MCI data, respectively. 

Here, experimental results are presented. An SVM classification algorithm with a linear kernel was used for Alzheimer’s disease diagnosis. The evaluations were done using only one set of reduced data, and then using different numbers of repetitions of the reduced data. Toward this aim, 144 selected features were used (including 132 MRI voxel and volume values, 1 MMSE, 4 types personal information, 3 cerebro-spinal fluid biomarkers, and 4 PET image voxel sizes). Table 2 shows the accuracy of discriminating AD, MCI, and NC from each other using each group of data, alone.

As deduced from Table 2 and Figure 5, Figure 6 and Figure 7, the AUC values increased after repeating reduced features each time until the fifth repetition. As can be seen in Figure 5, using repeated features in width (emphasized features) compressed box-plots further, demonstrating higher stability in the classification. The bold numbers in Table 2 show the highest values for accuracy and other performance measures for the proposed method.

In addition, we have added more explanatory experiments as Appendix A. There, we compare Homogeneous and Heterogeneous Emphasis Learning methods. The Homogeneous Emphasis Learning repeats all the selected features equally and homogeneously, and the Heterogeneous Emphasis Learning repeats selected features unequally and heterogeneously.

## 5. Discussion 

As mentioned in the main idea of this paper and as is clear in Table 3, this method had lower effects on models with low rates of accuracy and fairly weak models. In other words, this method emphasizes very strong features. As can be seen in Table 2, regarding the models in which there was a reduction in performance after dimension reduction, or where there was not much positive change, repetition in dimensions could not cause a considerable increase in the model performance. This is also predictable considering the main idea of the method. Because the model emphasizes valuable features, when the extracted features do not have any considerable effect on model performance, repeating them cannot be very helpful in increasing the model performance. Issues regarding the main idea and the results of incorporating it are discussed in the following.

### 5.1. Feature Representation

Across classification tasks, different numbers of input features can affect AD diagnosis in supervised learning. In the literature, the effects of considering different input sizes for different classification problems have been extensively discussed. The original features are informative for brain disease diagnosis, but this increase in the feature vector size will result in a better and more calculable diagnosis. 

In comparison with the original features, the proposed method greatly improved the diagnostic accuracy for all the considered classification problems. The proposed method outperformed the other methods in three binary classification problems. Using this method, notwithstanding the limited number of samples, helped reduce errors for classification problems and hence enhanced the classification accuracy. Previous methods can only use limited number of features in learning, but this limitation is overcome here.

There will not be an interpretation of the trained model or the feature representations. Each added unit in the input represents a linear combination of the high-level input features. That is, by repeating each high-level feature (e.g., mean intensity from FDG-PET or GM volume from MRI), the model can cover different relations for low-level features. Using this method, and from a neuro-scientific perspective, the relations from MRI features and from FDG-PET features could be enhanced. This way, new and increased inputs of the high-level features represent their helpfulness in classifying patients and healthy normal controls. Using this method cannot help us to interpret or visualize the model’s outputs, and this remains one of the unsolved pattern recognition and machine learning problems. In contrast, it is clear that this combined information will be useful in AD/MCI diagnosis.

### 5.2. Feature Reduction and Increasing—Feasibility of the Proposed Method

Here, we compare the results of the proposed method with PCA and the before-PCA results. In the feature set, we considered the clinical labels and clinical scores of MMSE. We observed that the method using increased specific numbers of feature packets outperformed others (Table 3). Here, we selected 15 to 30 PCA components explaining approximately 94.5% to 100% of the variance to test the proposed method.

The reason for the high performance of the proposed method can be explained as follows.

Consider learning perfect objects; when repeating them in the training process, the model can learn the object’s features better and better. This is because the richer information of the object can be learned by the model. Similarly, when we repeat perfect features (features obtained after applying PCA that gave us rich object features), the model can learn richer information about the objects. Similar to repeating objects after a specific number of repetitions, the model can be over-trained, and specifying the number of the repetitions is a precise action. Therefore, the method can make features that can accurately model the target values (i.e., labels and clinical scores and imaginary symptoms). This is the definition of repeating features in width (feature repetition or emphasized features) instead of length (sample repetition) in the algorithm. We can say the repeated features for the labels could discriminate AD and MCI patients from NC subjects.

As we said, the main idea of this paper is to repeat the most efficient features in classification. Our experiment showed that the theory of finding and repeating the good features of the data for classification problems will work excellently (Table 2 and Table 3).

### 5.3. Classification Algorithm

SVM is a widely used algorithm in the area of Alzheimer ’s disease. We selected this algorithm with a linear kernel. Non-linear SVMs usually achieve better performance, but here in our tests, performance differences between the two types of kernel were not significant, and linear SVMs were faster to train, as shown in Table 4.

As can be seen in Table 5, using all the data for the AD vs. NC task took 5.5170 seconds and after PCA it took 4.5012 s, increasing accuracy (from Table 2: accuracy increased from 95.54% to 97.20%) while reducing time. For this task, with five-fold repetition, the time was 5.9490 s while the accuracy was 98.81% (i.e., slight increase in the time complexity and a fair increase in the performance).

### 5.4. Comparison with the State-of-the-Art Methods 

To validate the performance of the proposed method, we present the significance of the results in Table 2. The performance of our proposed method is compared to other state-of-the-art methods in Table 3. For comparison, we used all 705 ADNI samples. The accuracy rates of our proposed method were 98.81%, 81.61%, and 81.40% for AD vs. NC, AD vs. MCI, and MCI vs. NC classification, respectively.

Most of the listed methods in Table 3 used the ADNI database, and we used all the images and data in ADNI, consisting of 705 participants. This is in contrast to some of the reported methods that used a portion of the ADNI samples. When the proportion of the used sample was sufficient compared to the entire sample, we compared our proposed method to theirs. As can be seen in Table 3, the proposed method obtained the highest accuracy (98.81%) in AD vs. NC diagnosis, and due to the balanced dataset, the accuracy is a suitable performance measure.

Additionally, the sensitivity (recall rate) of the proposed method (98.52%) was the highest, as can be seen in Table 3. The proposed method stood in second place with a specificity of 99.21% against the perfect detection rate of Ben Ahmed et al. [51]. However, Ben Ahmed et al.’s method had a high false alarm rate while the proposed method succeeded in achieving a trade-off between the two. Note that lower specificity of our method can be cover by higher sensitivity, e.g., 94.83% of our method vs. 87% in the case of Ben Ahmed et al.’s method. Nevertheless, combined methods can be used to achieve better performance.

In [23], 70 MRI images from the ADNI dataset were used for AD vs. NC classification—35 images of AD patients and 35 images of NC subjects. They achieved 100% accuracy in distinguishing between the two groups. Because the number of subjects used in our study was 10 times greater than this last study, it is not fair to compare this study with our proposed method. [18] achieved 89.66% accuracy in binary AD classification (recognizing AD from subjects) but they achieved 92.85% accuracy in multiple classification using transfer learning. Using only MRI data [24] could classify AD vs. NC with an accuracy rate of 93.01%. They achieved this performance using only 186 MR images. Using 785 MRI data [25] classified progressive MCI vs. static MCI with an accuracy of 86%, using combined MRI, APOe4 genetic data, and available clinical practice variables and cognitive measures including neuropsychological cognitive assessment tests like the Dementia Rating Scale (CDRSB), the Alzheimer’s Disease Assessment Scale (ADAS11, ADAS13), episodic memory evaluations in the Rey Auditory Verbal Learning Test (RAVLT).

We could redress some imperfections of the proposed method using another method that has better performance. According to Table 3, in most of the performance measures the proposed method was in the first or second place, and we can see that the proposed method was dominant, and had the highest accuracy compared to other methods.

### 5.5. Limitations of the Work 

The proposed method has some limitations. In PET imaging, the partial volume effect induced by a combination of image sampling and the restricted resolution of PET in the reconstructed images can bring under- or over valuation for regional radioactivity condensation. Therefore, more errors in statistical parametric images may occur [54]. Here, we must say that we could not download relevant PET images from ADNI and we only downloaded the extracted values of PETs from the ADNI database. 

Combination of multiple tissue values would likely affect the differences between voxels of gray and white matters. Since our method is ROI-based feature selection, this partial voxel quality reduction would have an inconsiderable effect on the performance of the method.

We can say the proposed structure used to form the feature sets in this experiment could be non-optimal for other datasets. We need studies such as those learning optimal and strong feature sets for repetition and practical use of the proposed method. The NC group in the dataset could include both healthy controls and subjective cognitive complaints because there is no supplementary information about this group. The features concatenation from MRI, FDG-PET, MMSE scores, and CSF modalities into a single vector and repetition of the features after the feature reduction could efficiently distinguish between AD and MCI patients and NC subjects.

## 6. Conclusions

In this study, we proposed a simple but practical and effective method for classification, and tested it for Alzheimer’s disease diagnosis. Our proposed method found the best features and repeated them until no further improvements to classification performance were obtained. We examined our method on the ADNI database of AD. The experiments showed that we could achieve much better performance using the combined features of MRI, MMSE, and personal information, especially when we repeated the reduced features on all three binary classification problems (i.e., AD vs. NC, AD vs. MCI, and MCI vs. NC). Experiments indicated the performance and effectiveness of the proposed method: accuracy rates of 98.81%, 81.61%, and 81.40% for AD vs. NC, AD vs. MCI, and MCI vs. NC classification problems, respectively. As can be seen, using this method increased the performance of the three binary problems incredibly. The results showed that the classification accuracy was improved with the optimized feature selection, which indicates that the information gain method can be used to select the more sensitive anatomical regions in AD and MCI diagnosis. Using other feature reduction or selection methods and repeating reduced data could be the subject of a future work. Combining the results of other feature reduction and selection methods and establishing a classification framework, while using them, could be another future work. This study employed VBM8 tools, which yielded promising results. However, there are robust segmentation tools to explore (e.g., CAT12) that could be used to improve the diagnosis results. As another future work, applying this method to clustering can be recommended. Finally, features recommended by experts could be put through the proposed model in order to achieve better performance as a future work.

## Figures and Tables

**Figure 1 sensors-20-00941-f001:**
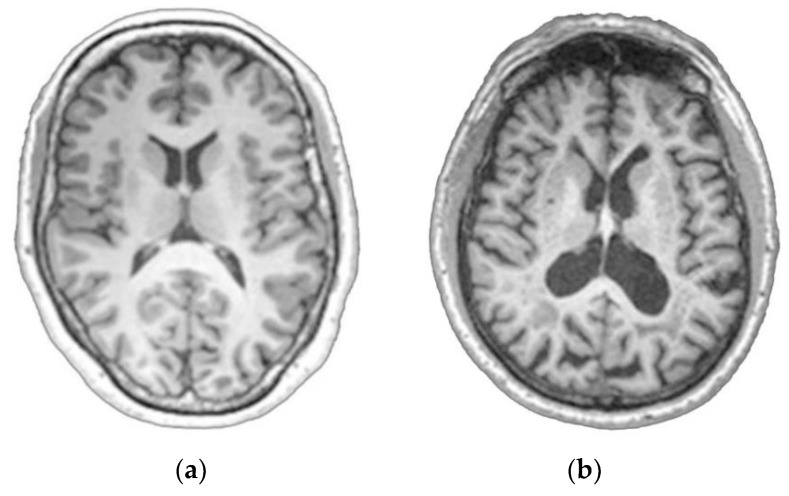
MRI imaging sample: (**a**) NC subject; (**b**) AD patient.

**Figure 2 sensors-20-00941-f002:**
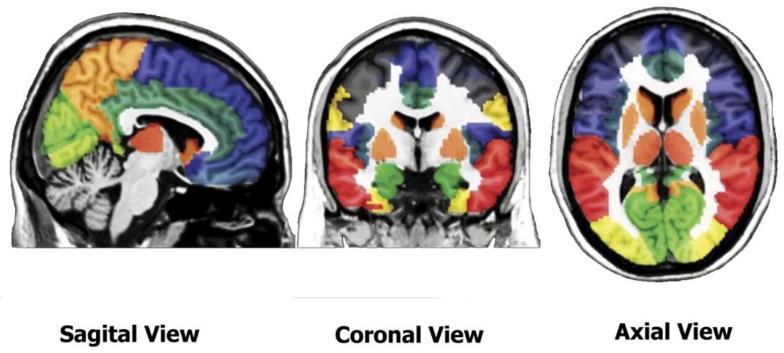
Automatic Anatomical Labeling (AAL) atlas. Different regions of brain are in different colors.

**Figure 3 sensors-20-00941-f003:**
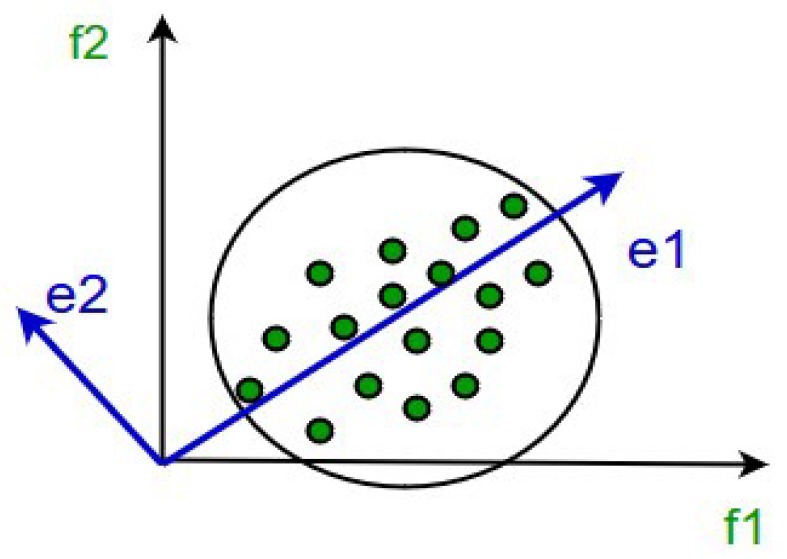
Schematic representation of dataset eigenvectors.

**Figure 4 sensors-20-00941-f004:**
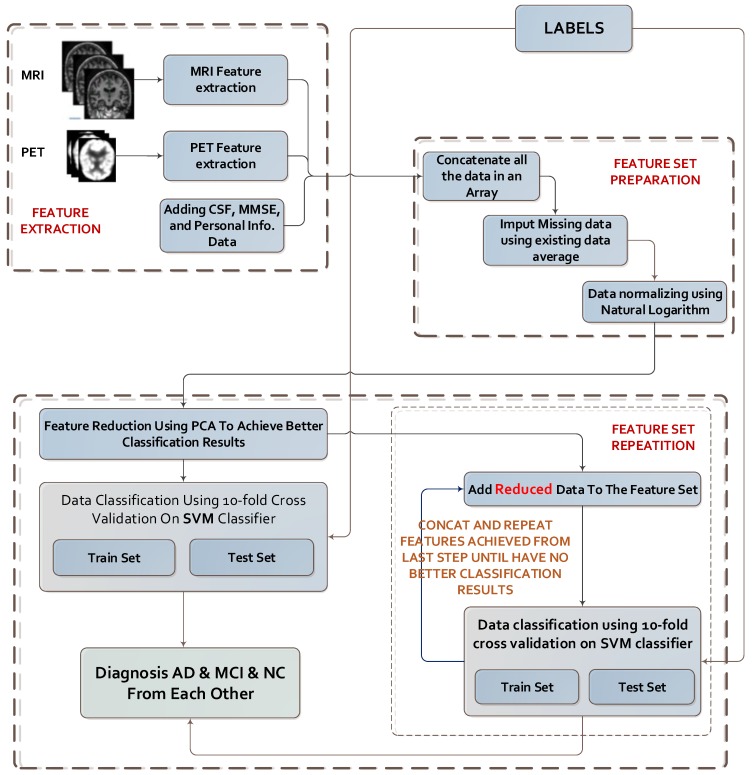
Diagram of steps of the proposed method (Emphasis Learning). CSF: cerebro-spinal fluid.

**Figure 5 sensors-20-00941-f005:**
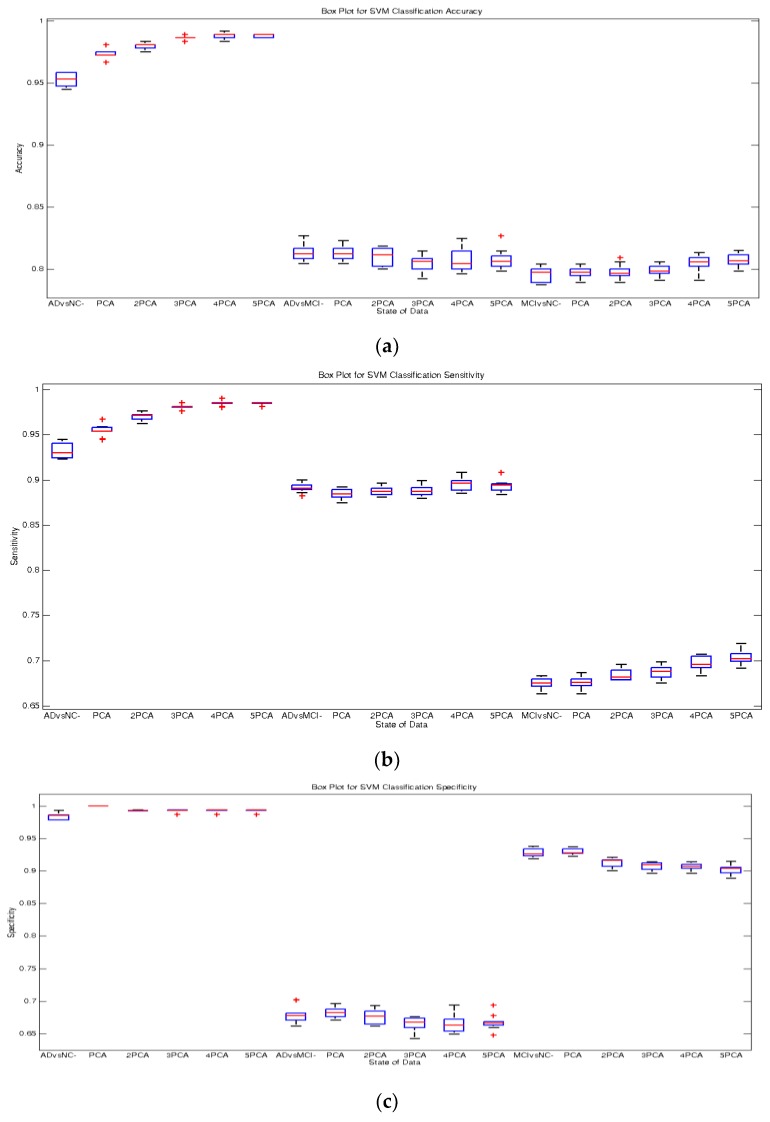
Boxplots for the recognition of AD, MCI, and NC subjects: (**a**) accuracy, (**b**) sensitivity, and (**c**) specificity of all features and 25 PCA elements for 1–9× repeated feature reduction.

**Figure 6 sensors-20-00941-f006:**
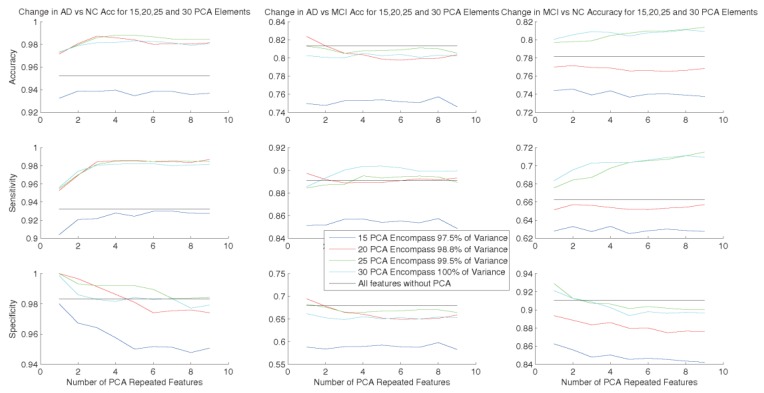
Comparison of changes in average accuracy, sensitivity and specificity for 15, 20, 25, and 30 PCA elements in 1–9× repeated feature reduction, and all the features (black line).

**Figure 7 sensors-20-00941-f007:**
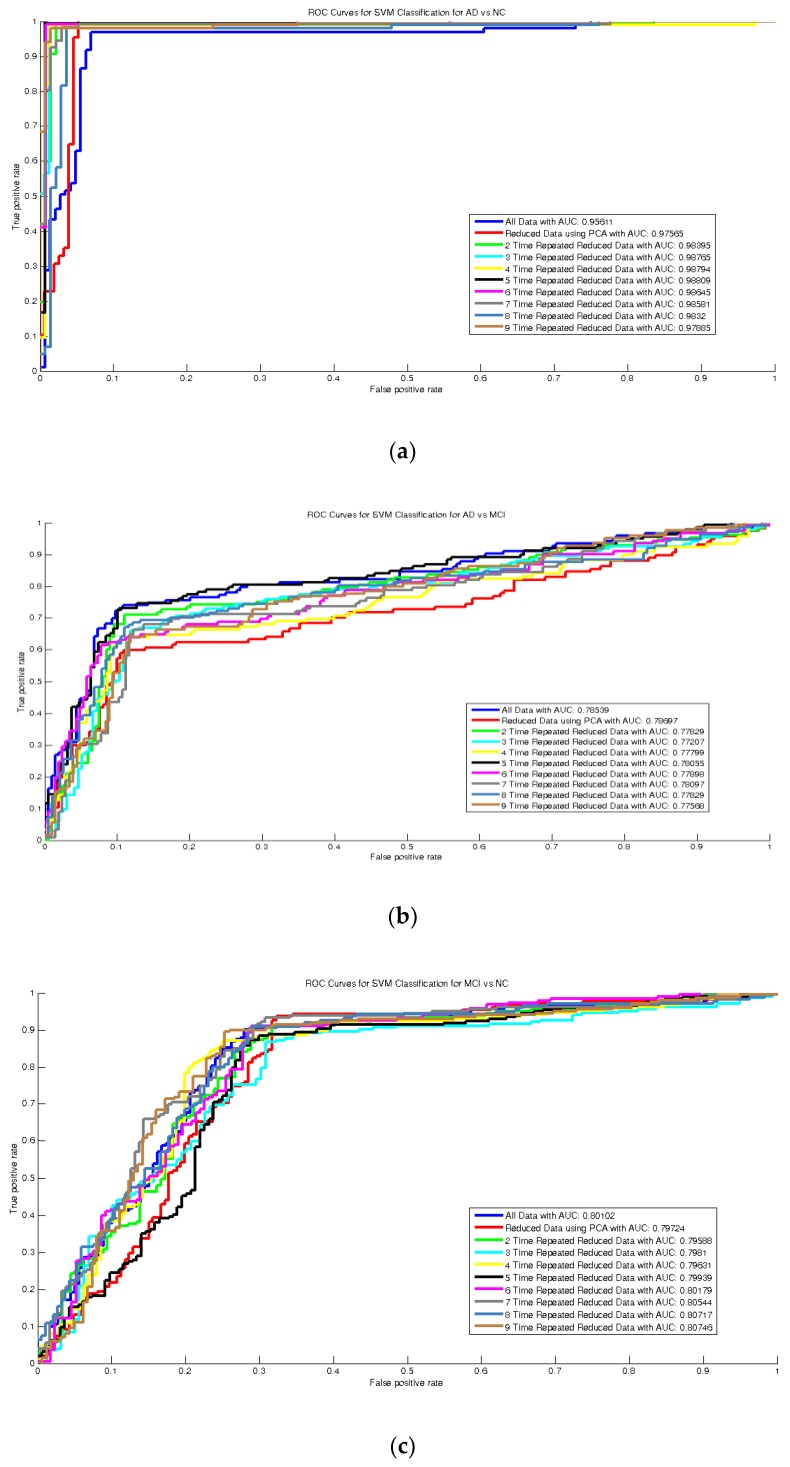
Receiver operating characteristic (ROC) curves for recognition of: (**a**) AD vs. NC, (**b**) AD vs. MCI, and (**c**) MCI vs. NC, of all features and 25 PCA elements for 1–9× repeated feature reduction.

**Table 1 sensors-20-00941-t001:** Summary of demographic data of patients and subjects. AD: Alzheimer’s disease; MCI: mild cognitive impairment; MMSE: Mini-Mental State Examination; NC: normal control.

	Count	Male	Female	Married	Widowed	Divorced	Never Married	Average Age	Average MMSE
**AD**	156	76	80	127	18	8	3	74.89	23.32
**NC**	211	110	101	142	38	17	14	75.91	29.13
**MCI**	338	215	123	269	39	24	6	74.51	27.05
**Total**	705	401	304	538	95	49	23	75.01	26.85

**Table 2 sensors-20-00941-t002:** Comparing performance metrics. Classification accuracy (ACC), sensitivity (SEN), specificity (SPE), positive predictive value (PPV), negative predictive value (NPV), and area under the curve (AUC) for all features and 25 principal component analysis (PCA) elements.

Data	Classes	ACC (%)	SEN (%)	SPE (%)	PPV (%)	NPV (%)	AUC
**All Data**	AD–NC	95.54	93.74	98.32	98.84	91.09	0.9577
	AD–MCI	81.41	89.02	68.09	82.99	78.00	0.7835
	MCI–NC	79.41	67.48	92.37	90.56	72.34	0.7993
**Reduced Data Using PCA**	AD–NC	97.20	95.46	**99.86**	**99.90**	93.53	0.9768
	AD–MCI	**81.61**	88.45	**69.02**	**84.03**	76.39	**0.7846**
	MCI–NC	79.45	67.20	**92.96**	**91.33**	71.97	0.8011
**2 × Reduced Data**	AD–NC	98.03	97.18	99.26	99.47	96.09	0.9831
	AD–MCI	80.37	88.57	66.38	81.80	77.28	0.7766
	MCI–NC	79.94	68.49	91.64	89.31	74.03	0.7991
**3 × Reduced Data**	AD–NC	98.61	98.15	99.27	99.47	97.46	0.9863
	AD–MCI	80.47	88.90	66.26	81.62	77.98	0.7767
	MCI–NC	79.93	68.70	90.80	87.85	74.98	0.7980
**4 × Reduced Data**	AD–NC	98.67	98.24	99.27	99.47	97.59	**0.9876**
	AD–MCI	80.61	88.92	66.59	81.81	78.02	0.7784
	MCI–NC	80.55	69.84	90.62	87.47	76.20	0.7998
**5 × Reduced Data**	AD–NC	**98.81**	98.52	99.21	99.42	97.98	**0.9875**
	AD–MCI	80.69	89.46	66.37	81.29	79.39	0.7803
	MCI–NC	80.92	70.64	90.17	86.60	77.36	0.8016
**6 × Reduced Data**	AD–NC	98.59	98.51	98.69	99.03	97.98	0.9866
	AD–MCI	80.81	89.47	66.65	81.46	79.43	0.7793
	MCI–NC	80.67	70.26	90.05	86.43	77.05	0.8045
**7 × Reduced Data**	AD–NC	98.50	**98.61**	98.37	98.80	**98.11**	0.9852
	AD–MCI	80.71	89.07	66.66	81.82	78.32	0.7778
	MCI–NC	**81.44**	71.28	90.54	87.09	77.89	0.8056
**8 × Reduced Data**	AD–NC	98.34	98.56	98.04	98.56	98.04	0.9835
	AD–MCI	80.84	**89.55**	66.57	81.45	**79.52**	0.7789
	MCI–NC	81.42	**71.51**	90.30	86.84	**77.98**	0.8075
**9 × Reduced Data**	AD–NC	98.31	98.41	98.18	98.65	97.85	0.9822
	AD–MCI	80.51	88.91	66.43	81.62	78.13	0.7767
	MCI–NC	81.40	71.22	90.54	87.09	77.82	**0.808**

**Table 3 sensors-20-00941-t003:** Comparison of the proposed method with other methods; including dataset and indicators. EL: Emphasis Learning. ADNI: Alzheimer’s Diseases Neuroimaging Initiative.

Method	Data type(s) (n, Dataset)	AD vs. NC				AD vs. MCI				MCI vs. NC			
		Acc%	Sen%	Spec%	AUC	Acc%	Sen%	Spec%	AUC	Acc%	Sen%	Spec%	AUC
**Zhang et al., 2011 [45]**	MRI, PET, CSF, MMSE, ADAS-Cog (202, ADNI)	93.20	93.00	93.30	0.98	-	-	-	-	76.40	81.80	66.00	0.81
**Dai et al., 2013 [46]**	MRI (83, OASIS)	90.81	92.59	90.33	0.94	85.92	82.46	87.59	0.87	81.92	78.51	88.34	0.81
**J. Liu et al., 2016 [47]**	MRI, PET (710, ADNI)	94.65	95.03	91.76	0.95	88.63	91.55	86.25	0.91	84.79	88.91	80.34	0.83
**Beheshti et al., 2017 [24]**	MRI (186, ADNI)	93.01	89.13	96.80	0.935	-	-	-	-	-	-	-	-
**Mishra et al., 2018 [48]**	MRI (417, ADNI)	89.15	85.06	92.53	0.93	-	-	-	-	-	-	-	-
**Khedher et al., 2015 [49]**	MRI (818, ADNI)	88.96	92.35	86.24	0.93	84.59	88.75	83.07	0.89	82.41	84.12	80.48	0.81
**Lian et al., 2019 [50]**	MRI (1457, ADNI)	90.00	82.00	97.00	0.95	-	-	-	-	-	-	-	-
**Ben Ahmed et al., 2014 [51]**	MRI (218, ADNI)	87.00	75.50	100	0.85	72.23	75.00	70.00	0.76	78.22	70.73	83.34	0.77
**Zhou et al., 2018 [52]**	MRI (507, ADNI)	93.75	87.5	100	-	-	-	-	-	-	-	-	-
**Suk et al., 2014 [53]**	MRI, PET, CSF, MMSE, ADAS-Cog (202, ADNI)	93.05	90.86	94.57	0.95	88.98	82.11	90.65	0.90	83.67	96.79	57.28	0.82
**Maqsood et al., [18]**	MRI (392, OASIS)	89.66	100	82	-	-	-	-	-	-	-	-	-
**Proposed Method (EL)**	**MRI, PET, CSF, MMSE** **(705, ADNI)**	98.81	**98.52**	99.21	0.987	81.61	88.45	69.02	0.785	81.40	71.22	90.54	0.81

**Table 4 sensors-20-00941-t004:** Time complexity of SVM with linear kernel vs. Radial Basis Function (RBF) kernel.

Data	Classes	SVM Training Time—Linear Kernel (s)	SVM Training Time—RBF Kernel (s)	Linear—RBF (S)
**All Data**	**AD–NC**	5.5170	5.7599	−0.2429
	**AD–MCI**	6.7963	9.6903	−2.8941
	**MCI–NC**	6.8982	9.0480	−2.1498
**Reduced Data Using PCA**	**AD–NC**	4.5012	5.2301	−0.7289
	**AD–MCI**	5.7727	6.6944	−0.9217
	**MCI–NC**	7.3574	9.5005	−2.1432
**3 × Reduced Data**	**AD–NC**	5.0067	7.8505	−2.8437
	**AD–MCI**	6.3709	9.9687	−3.5978
	**MCI–NC**	8.1439	12.9038	−4.7599
**5 × Reduced Data**	**AD–NC**	5.9490	9.6567	−3.7077
	**AD–MCI**	7.3554	12.8005	−5.4451
	**MCI–NC**	11.9281	18.2044	−6.2763
**9 × Reduced Data**	**AD–NC**	8.7938	12.0661	−3.2722
	**AD–MCI**	10.0609	16.9106	−6.8497
	**MCI–NC**	14.0926	23.2651	−9.1726

**Table 5 sensors-20-00941-t005:** Accuracy of AD, MCI, and NC classification, using each group of data, alone.

CLASSES	AD–NC ACC%	AD–MCI ACC%	MCI–NC ACC%
**Personal Information**	0.609	0.595	0.553
**MMSE Data**	0.919	0.785	0.703
**MRI Data**	0.868	0.684	0.697
**CSF Data**	0.594	0.524	0.643
**PET Data**	0.625	0.667	0.574

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
