# Peer review of "Emphasis Learning, Features Repetition in Width Instead of Length to Improve Classification Performance: Case Study—Alzheimer’s Disease Diagnosis"

_sensors, 2020, doi:10.3390/s20030941_

Round 1
Reviewer 1 Report
In this paper, the authors used a set of features achieved from MRI, PET and clinical assessment for AD classification. The PCA technique was used to improve the prediction accuracy.
Main concern:
Concatenating the MRI features with PET and clinical information has been fully investigated in a series for AD classifications. See the following papers. Besides, PCA is an old fusion data reduction technique. Consequently, I do not see a novelty in this study. What we can learn from this study which is not investigated yet?
Zhang, Daoqiang, Dinggang Shen, and Alzheimer's Disease Neuroimaging Initiative. "Multi-modal multi-task learning for joint prediction of multiple regression and classification variables in Alzheimer's disease." NeuroImage 59.2 (2012): 895-907.
Gray KR, Aljabar P, Heckemann RA, et al. Random forest-based similarity measures for multi-modal classification of Alzheimer’s disease. Neuroimage 2013;65:167–75.
Other comments:
The author stated that “PCA can efficiently be used to extract more efficient features of data and enhance classification accuracy.” I am not completely agree with this claim. In fact, PCA is a data reduction technique which maps input data into a smaller space. Although sometimes PCA can improve the prediction accuracy, but it is not a significant improvement. See table 2 in your paper. for AD/MCI task the classification before PCA was 81.41% and after applying the PCA 81.61%. Moreover, the dimensionality of features is not super higher than number of samples in this study. So, using PCA to avoid of “course of dimensionality” does not make a strong sense here.
Regarding the MRI pre-processing, it is strongly recommend that use CAT12 instead of VBm8 as it provides a more robust segmentation to trace the morphometric changes.
Farokhian, Farnaz, et al. "Comparing CAT12 and VBM8 for detecting brain morphological abnormalities in temporal lobe epilepsy." Frontiers in neurology 8 (2017): 428.
Regarding the MRI pre-processing, it is not clear that the segmentation phase is done using SPM or FSL?
Details of PET pre-processing should be reported. The authors reported 132 MRI features. The authors should clarify the details of MRI features extraction. How many features per region? Regarding PET imaging, why only 4 features? See the following paper as an example for PET data extraction:
Zhang, Daoqiang, Dinggang Shen, and Alzheimer's Disease Neuroimaging Initiative. "Multi-modal multi-task learning for joint prediction of multiple regression and classification variables in Alzheimer's disease." NeuroImage 59.2 (2012): 895-907.
The authors processed WM images whereas only GM images were used!
The authors claimed that “Increasing Dimention of data to achieve better classification results”. Again I am not agree with this point. If the authors plot the prediction accuracy as a function of all number of PCs (Fig 4), they will see that after a specific number it is not always increasing! Indeed, it will be decreasing and increasing due to noise or measurement errors .
The authors concatenated (MF, PF, CD, PI, MS). From clinical view, it is not clear that which set of features is more informative for AD classification. Does concatenating data contribute to a more robust CAD?
Figure 2, “Imputing Missing data using existing data average”. What is the aim of this step? And why? For example, for an AD patient, the MMSE score is missed. How does your procedure compute it?
Table 2, the authors used 25 PCs and changed the data set to achieve maximum accuracy followed by a 10 fold cross-validation. For AD/HC the maximum ACC achieved after 5 repetitions and for AD/MCI in the first one. Please note that it would be nice to find the parameters using an automatic procedure valid for all tasks. Also, I strongly recommend adjusting parameters only based on the training set in each iteration.
Table 3, it is recommended to compare your study with recent methods in the field (i.e., techniques after 2015)
The paper should be edited by an English proofing service.
Reviewer 2 Report
Briefly, describe proposed approach in abstract. ‘’PCA was introduced by Karl Pearson for the first time’’. Give reference with date. Figure 1 is generic. Provide example from real AD MRI. It is not clear if Xnorm is used or Xstd is used for training classifier? The box containing steps of the algorithm is incomplete, see the last step is it is clearly shown within the box. Why linear SVM was used? Why not a nonlinear SVM? Discuss required processing time of the proposed approach. Discuss feasibility of the proposed approach. Cite very recent works where high accuracy was achieved: Performance of machine learning methods applied to structural MRI and ADAS cognitive scores in diagnosing Alzheimer’s disease, Biomedical Signal Processing and Control 52, 414-419, 2019. Image characterization by fractal descriptors in variational mode decomposition domain: application to brain magnetic resonance, Physica A: Statistical Mechanics and its Applications 456, 235-243, 2016.
Round 2
Reviewer 1 Report
I would like to thank the authors for addressing several of the comments made in my previous review. I feel that there are pints, which should be better addressed/explained in the manuscript.
Item 1 – Regarding using CAT12 at the pre-processing stage, the authors persisted to report results based on VBM8. However, it recommended using CAT12 for further studies. In addition to the added sentence, it would be nice to include the related reference.
https://www.frontiersin.org/articles/10.3389/fneur.2017.00428/full
Item 2- The authors stated that “The PET images were aligned to the corresponding MR images strictly. Recognized areas from MRI Gray Matter tissue volume, and average voxel values, and average of PET voxel values, were used as features”. This sentence means that authors had access to PET images, pre-processed the PET images and extracted metabolic features from PET images. This sentence is in contrast with the next sentence that “Since we could not access PET Images in ADNI, we used only 4 PET features that were extracted and uploaded to the ADNI database”. I am surprised that the authors have no access to download PET images from ADNI. Many research groups download PET images from ADNI as well as MRI images. I think the authors should modify the respective section and provide more details about PET features ( type, unit and …) for readers. For example, we know that the GM voxel signal intensity has been used as MRI features. Thus, the PET section is not clear to me.
Item 3 – The authors estimated the missing value in PET and CSF using available data. Personally, I am not sure about the accuracy of this procedure. If there is a reference for this case, please add it.
Item 4- As I compare table 3 in the new version and old version, both are the same without adding most recent papers! I would suggest to include these papers as new techniques in AD classification studies not only into table 3 but also in the discussion section.
https://www.sciencedirect.com/science/article/pii/S0010482517300483
https://www.sciencedirect.com/science/article/abs/pii/S105381191930031X
https://www.mdpi.com/1424-8220/19/11/2645/htm
Item 5- As for Table 3, I think it is not a fair comparison. The authors should add the data used in each study in that table. For example, your ACC is reported based on MRI+PET+CSF whereas other studies reported based on only MRI information.
Item 6- Figure 4, “MRI Feature extraction” should be “PET Feature extraction” for PET images”.
ITEM 7- Please recheck all Abbreviations.
Item 8- The quality of the figures can be improved.
